# A Multi-Pumping Gradient Calibration Module for Potentiometric Determination of Nitrate in Surface Water

**DOI:** 10.3390/molecules28020493

**Published:** 2023-01-04

**Authors:** Sławomir Kalinowski, Paweł Kościelniak, Elwira Wierzbicka, Stanisława Koronkiewicz

**Affiliations:** 1Department of Chemistry, University of Warmia and Mazury in Olsztyn, 10-957 Olsztyn, Poland; 2Faculty of Chemistry, Jagiellonian University, 30-387 Krakow, Poland

**Keywords:** flow analysis, solenoid micro-pump, calibration, ion selective electrode, nitrate

## Abstract

The novel, automated, multi-pumping flow system (MPFS) for online calibration and determination of nitrate in surface water is presented for the first time. The system was equipped with micropumps of three different nominal volumes (10, 20, and 50 µL). As a result, it was possible to prepare from one standard, directly in a flow system, up to seven standard solutions. Determination of nitrate was conducted in stop-flow conditions and is based on a commercially available ion selective electrode (ISE) application. It was found that the linearity and slope of the calibration graphs depend mainly on the characteristics of the ISE. The obtained results were very repeatable, owing to the high precision of the micro-pumps used. The R.S.D. for the stroke volume of each micro-pump was below 1%. The accuracy of the method was checked through determination of nitrate in surface water samples. The obtained results were compared with those of the reference method (photometric Hach cuvette tests). It was found that, at a 96% confidence level, the difference between the results obtained by the proposed method and the reference method was statistically insignificant. The accuracy of the method was confirmed through the determination of nitrate in Certified Reference Material. The relative deviation (R.D.) of the measured and the certified concentrations was 5%.

## 1. Introduction

Nitrogen is an essential nutrient for primary production in aquatic ecosystems. Excessive nitrogen concentration in the waters leads to the unfavorable eutrophication phenomenon. Increases in nitrogen concentrations in surface water are primarily derived from such anthropogenic sources as agricultural activity, domestic and industrial wastewater discharges [1]. Nitrates are the most common forms of nitrogen to be found in aquatic systems. In waters with sufficient dissolved oxygen concentration, ammonium is transferred to nitrate, for which nitrite is an intermediate species (i.e., NH_4_^+^ → NO_2_^−^ → NO_3_^−^). Therefore, the determination of nitrate is vital to understand its biogeochemistry in aquatic systems and its role in eutrophication. Our need to monitor these ions is unquestionable.

Flow techniques have proved to be suitable tools for the in-situ determination of nutrient species in natural waters since these techniques are fast, portable, easy to automate, and they provide high-quality data with high temporal resolution [2]. In many modern applications (e.g., HPLC, capillary electrophoresis, or hyphenated methods), controlled flow techniques are welcome and widely used because they help to improve the precision and reproducibility of the measurements [3,4]. Most reported flow applications of nitrate determination are based on flow injection analysis (FIA) [5,6,7]. Other flow methods have been developed as well, such as sequential injection analysis (SIA) [8], multi-syringe flow analysis (MSFA) [9], and multi-pumping flow analysis (MPFA) [10,11].

Despite its many advantages, the stage of standard solution preparation remains the Achilles’ heel of flow techniques. According to the flow analysis idea, this step should also be automated. The conventional, interpolative “calibration curve method” is commonly used in analytical practice. For this purpose, the special units dedicated to controlled dilution of a single standard, like mixing chambers, fully rotary valves, or systems of dilution loops, are used [12,13]. These techniques are based on controlled, precise dispersion, which occurs in flow systems [14]. Similar possibilities are offered by the use of solenoid-operated micro-pumps in MPFA systems. Micro-pumps play a similar role as micropipettes in stationary conditions. They are responsible for the precise dosing of solutions. MPFA systems for online dilution are also regularly reported in analytical publications [11,15,16,17,18]. In all of these systems, the solutions, after dosing, must be mixed. It is performed in a proper length mixing coil.

For nitrate determination in natural water, spectroscopic detections are the most widely used [19,20]. The most common approach to the detection of nitrite is spectrophotometry based on the Griess reaction. The nitrates are determined after their reduction to nitrite. Spectrofluorimetric, chemiluminescence, and electrochemical detection are also mentioned, additionally, in combination with chromatography or capillary electrophoresis. All of these methods, sometimes very sophisticated and expensive, use a lot of reagents and come with their own sets of advantages and limitations.

One of the common approaches is based on the use of ion-selective electrodes (ISEs) [21,22]. ISEs are simple to use, and do not necessitate additional, expensive, and environmentally harmful reagents. One of the problems in conventional potentiometric measurements with ISEs is solution handling. Experiments using ISEs are normally done manually and are, therefore, rather time-consuming. For commercially available ISEs, checking the calibration every two hours is recommended. A new calibration curve should be prepared daily. Electrode measurements reproducible to ±2% can be obtained if the electrode is calibrated every hour [23]. One practical way to solve the problems with solution handling is to use flow analysis instrumentation. Examples of flow systems with fully automatic calibration and potentiometric detection using ISEs can be found in the available literature. Fernandes et al. [24] developed a flow setup based on the sequential injection analysis providing off-line and in-line calibration, ion-selective electrode characterization, standard addition techniques, and titration procedures that could be carried out without any stock solutions conventional handling. The system was designed for routine analysis of vitamin B6 in pharmaceuticals. Another example is a computer-controlled system for ISEs automatic calibration that was tested in the determination of Na^+^ ions in batch, steady-state, and flow injection measurements [25]. The aim of this study was to develop a simple, automated flow system and procedures allowing for calibration and direct determination of nitrate in natural water. To the best of our knowledge, a flow system with auto-calibration based on solenoid micro-pumps for the potentiometric ISE determination of nitrate ions has not been developed so far. The research was conducted towards the application of the developed procedure in monitoring surface water pollution.

## 2. Results and Discussion

### 2.1. Re-Calibration of Micro-Pumps

As a part of the preliminary tests, the volume of the solenoid micro-pumps used has been checked and recalculated. The need to recalculate the nominal stroke volume indicated by the micro-pumps has been pointed out elsewhere [10,26]. For this purpose, water was pumped at the maximum frequency allowed by the manufacturer (2 Hz). The pumped water was collected and weighed. Ten injections of water were taken for each weighing. The calibration was carried out in four repetitions. The real stroke volume was obtained by dividing the average weight by the density of the water. To check how much the real stroke volume depends on the backpressure, the experiment was conducted twice: Using a short (30 cm) tube at the pump outlet and using a 2 m long tube. The results of this experiment are summarized in Table 1. The data corresponding to the 2 m long tube are given in brackets.

As can be observed, the precision of the micro-pumps used was very good. R.S.D. for stroke volume of each micro-pump was below 1%. Unfortunately, the accuracy of the micro-pumps was low, especially for the smallest one. For pumps P3 and P6, having the nominal volume of 10 µL, the deviation of this volume from the real volume was equal, respectively, 31 and 43 percent. Additionally, it was observed that the real stroke volume showed no significant dependence on back pressure. Pumping accuracy decreases only slightly with the length of the tube placed at the outlet of the micro-pumps. Fortunately, the nominal volume of the micro-pumps, although significantly different from those declared by the manufacturer, was repeatable over time. Therefore, in the next research, when calculating the dilution of the standards in the flow system (Section 3.2.2), only the micro-pump volumes determined experimentally were used.

### 2.2. Calibration Graph

First, in order to check the correctness of the calibration with this method, the operation of the micro-pumps was properly programmed. Each calibration cycle consisted of six steps, in which six signals were registered, five for appropriately diluted standard and one for sample (see Section 3.2.2). The calibration cycle was repeated five times. The concentration of the standard solution was chosen so as to obtain, after dilution, a range near the expected concentration of analyte in the unknown sample. Four concentration ranges were checked, using dilution standards of concentrations 10, 50, 100, and 1000 mg L^−1^, as shown in Table 2. As a sample, solutions of known NO_3_^−^ ions concentrations were applied. All the calculations were made using log concentration values, which is typical when working with ISE.

During this experiment, the calibration graph equations, V = f(log CNO_3_), for successive standard concentrations (10, 50, 100, and 1000 mg L^−1^), n = 5, were as follows:y = −28.3x + 446, R^2^ = 0.905(1)
y = −43.5x + 465, R^2^ = 0.988(2)
y = −48.7x + 472, R^2^ = 0.996(3)
y = −51.1x + 472, R^2^ = 0.998(4)

The sensitivity of the presented method (the slope of the calibration graph) slightly increased with an increase of standard solution concentration applied and with the range of the calibration graph. It is a result of the lack of linearity of the ion-selective electrode characteristics in the studied concentration ranges. The coefficient of determination (R^2^) for the lowest concentration range studied (standard 10 mg L^−1^ dilution) was below expectations, but it is also related to the characteristics of the ISE applied [23], which is no longer straightforward at low analyte concentrations. Despite this, the accuracy of the determination was satisfactory. It was possible to correctly determine the concentration of NO_3_^−^ ions in the samples with a relative deviation (R.D.) no higher than 3%. Only for the lowest concentration, the accuracy is slightly worse, and the R.D. exceeds 7%.

An example of the calibration graph obtained for the standard concentration equal to 100 mg L^−1^ is presented in Figure 1 (blue color). The corresponding calibration graph obtained for the same ISE, but with standard solutions prepared manually, in batch conditions is presented in Figure 1 (yellow color). As can be observed, calibration graphs were characterized by a different slope. Fortunately, the determinations made by both methods, manual and flow, are very compatible (see Section 2.3).

### 2.3. Application to the Real Sample

The usefulness of the proposed method was tested through the analysis of several samples of water from lakes and ponds. The obtained results were compared with the results received using a manual, traditional procedure with the same ISE. A comparison with the reference, photometric method (the cuvette test LCK 339 (Hach-Lange)) was also performed. Cuvette tests are very popular nowadays in environmental analysis. The method has many advantages. One of them is the convenience and speed of performing the analysis. The spectrophotometers dedicated to cuvette tests are configured and pre-calibrated in the factory. Therefore, it is not necessary to calibrate the method every time when performing routine analyses. Nevertheless, the analysis time is still quite long (more than 15 min for one sample). Moreover, the analysis is not automated and requires fairly precise work from the analyst. Therefore, such tests, to our knowledge, are not used in environmental, automatic monitoring stations.

The obtained results are shown in Table 3. All the calculations were made using log concentration values. Since the nitrate content of the samples from the lakes and ponds varied considerably, different concentrations of the standards were used in constructing the calibration graphs. For the determination of NO_3_^−^ concentration in lake samples, the standard concentration used for calibration was 50 mg L^−1^ (log CNO_3_ = 1.70), in pond samples: 10 mg L^−1^ (log CNO_3_ = 1.00).

To establish whether the proposed method produces reliable results and whether those results are in agreement with the reference method, Student’s *t*-test was applied. First, we compared procedures utilizing the ISE methods: The proposed flow system and the manual procedure. It was found that the calculated t-value (0.378) was considerably lower than the tabulated t-value (t = 3.482, n = 3, *p* = 0.04). The calculated *p*-value was 0.731. This confirms the high compatibility of both procedures. When comparing the proposed flow system and reference cuvette tests, we found that the calculated t-value (3.334) was higher than the previous one but still lower than the tabulated t-value. The calculated *p*-value was 0.045. These results suggested that, at a 96% confidence level, the difference between the results obtained by the proposed flow method and the manual procedure, as well as the reference, the photometric method was statistically insignificant.

For evaluation of the accuracy of the proposed method, the concentration of NO_3_^−^ ions in Certified Reference Material (CRM) of river water (LGC 6025) was determined. The certified value of concentration for nitrate (as NO_3_^−^) in this sample was 38.0 mg L^−1^, with an uncertainty of 1.6 mg L^−1^. It means that the result is in the range from 36.4 to 39.6 mg L^−1^ (Log CNO_3_ = 1.58 ± 0.02). During the analysis of the CRM sample, four independent calibration graphs were prepared. For this purpose, the standard solutions at a concentration of 50 or 100 mg L^−1^ were diluted using the proposed system of micro-pumps. The comparison of measured results with the certified value was conducted according to the Application Note [27]. The method compares the difference between the certified and measured values with its uncertainty. After the measurement of the CRM, the absolute difference between the mean measured value (c_m_) and the certified value (c_CRM_) was calculated as:(5)$$\Delta_{m}=\left|c_{m}-c_{\mathrm{CRM}}\right|=\left|1.66-1.58\right|=0.08$$

The uncertainty of Δ_m_ is u_Δ_, which was calculated from the uncertainty of the measurement result (u_m_) and the uncertainty of the certified value (u_CRM_). As u_m_ the standard deviation of the mean measured value was used. The uncertainty u_Δ_ was calculated according to:(6)$$u_{\Delta }=\sqrt{u_{m}^{2}+u_{\mathrm{CRM}}^{2}}=\sqrt{0.04^{2}+0.02^{2}}=0.045$$

The expanded uncertainty U_Δ_, corresponding to a confidence level of approximately 95%, was obtained through the multiplication of u_Δ_ by a coverage factor (k) of 2:(7)$$U_{\Delta }=2\cdot u_{\Delta }=0.09$$

The results show that Δ_m_ < U_Δ_. Therefore, there is no significant difference between the measured results and the certified value. The relative deviation (R.D.) of the measured concentration (log CNO_3_ = 1.66) from the certified concentration (log CNO_3_ = 1.58) was 5%.

## 3. Materials and Methods

### 3.1. Chemicals and Reagents

All solutions were prepared with analytical-grade chemicals and using deionized water obtained from a Milli-Q Water Purification System (Millipore Corporation, Burlington, MA, USA) water purification system (resistivity > 18.2 MΩ cm). All reagents were of analytical grade. NO_3_^−^ working standard solutions were prepared by appropriate dilution of stock nitrate standard solution (Merck, Darmstadt, Germany), Cat. No. 1.19811.0500 with water and ionic strength adjuster (ISA, 2 mol L^−1^ (NH_4_)_2_SO_4_) obtained from Cole-Parmer (Vernon Hills, IL, USA), Cat. No. 27503-60. ISA was added to each standard and sample according to Cole-Parmer instruction manual [23]. All samples and standards were always kept and measured at the same temperature.

For the demonstration of the practical utility of the developed system, samples of water from lakes and ponds were collected in the region of Warmia and Masuria (Olsztyn, Poland), transported to the laboratory, and immediately used for nitrate determination without any treatment.

The cuvette tests LCK 339 (Hach-Lange, Loveland, CO, USA) were used as the reference method. The basis of this method is the spectrophotometric determination of nitrate ions, which, in solutions containing sulphuric and phosphoric acids, react with 2.6-dimethylphenol to form 4-nitro-2.6-dimethylphenol. A Certified Reference Material (CRM) of river water sample LGC 6025 (National Measurement Laboratory (NML) at LGC, Teddington, UK) was used for evaluation of the accuracy of the proposed method.

### 3.2. Apparatus

#### 3.2.1. Flow System

The flow manifold consisted of six solenoid-operated micro-pumps, flow lines, and a mixing chamber. The micro-pumps were purchased from Bio-chemValve Inc. (Boonton, NJ, USA) and Cole-Parmer (Vernon Hills, IL, USA) and have a nominal volume of 10, 20, or 50 µL. The flow lines were made of a PTFE tube (ID of 0.8 mm) and were obtained from Cole-Parmer. To decrease the time of analysis, the nominal volume of the pump used for sample injection was chosen to be 50 µL. A schematic diagram of the applying flow network is shown in Figure 2A.

The flow system was provided with four straight crosses, which for the convenience of use, have been combined and made in one block of Teflon, as shown in the photo (Figure 2B). The mixing chamber, of about 350 µL in volume, was also prepared in one block of Teflon. Inside the block, a cone-shaped top of the chamber was made. The inlets into the mixing chamber were drilled at its bottom and were oriented tangentially to the wall of the chamber. The outlet was positioned at the top of the chamber, which facilitates effective exchange of solutions in the chamber and removal of any gas bubbles. To encourage effective mixing, the solution of standard and diluent is injected into the mixing chamber opposite each other. A similar technical solution was used in the direct-injection detector of our design, described in an earlier publication [28]. Above the mixing chamber, there was a cylinder-shaped space of about 80 µL in volume. The ion-selective electrode was placed there. The inlet to this space was located on one edge of this space, the outlet on the opposite side, which facilitates effective exchange of solutions and removal of any gas bubbles, the same way it does in the mixing chamber. The inlet of the sample (done using pump P7) was located near the ISE, as was the inlet for the diluted or undiluted standard. This position seems to be advantageous from the point of view of its consumption and replacement.

The work of the entire system was PC-controlled by the measurement system developed specifically for flow analysis and was purchased from KSP Elektronika Laboratoryjna (Olsztyn, Poland) [29]. The hardware contained: Potentiostat: DPTG-217, DAQ-248; Programmable timer: TM-232; Valve controller: SNV-212; Pump controller: PPC-28. The software (KSP-‘Flow-potential’) was dedicated for potentiometric detection (read the potential of ion selective electrode) and controlling the solenoid micro-pumps and valves as well as the peristaltic pump. It allows realizing the flow analysis: injection, multicommutation and sequential. For the determination of nitrate ions, a combination ion-selective electrode was used (product no. 27504-22, Cole-Parmer, Vernon Hills, IL, USA). The full electrode characteristics and information on its reproducibility, concentration of possible interferences, temperature influences, etc., are included in the instruction manual [23].

#### 3.2.2. Procedure

All solutions were aspirated and then properly injected into the flow system by seven independent section solenoid micro-pumps. The pumps were responsible not only for the propelling of all solutions but above all, for precise dispensing. Rapid injection in counter-current forced a turbulent flow and facilitated efficient mixing of the solutions used [30]. An additional mixing coil, which can cause unnecessary dispersion of the reactants, is no longer needed.

For the calibration procedure, six solenoid micro-pumps were used (Figure 2, pumps P1 ÷ P6). The number of micropumps was chosen for the ability to select standard dilutions in the range from approx. 0.1 to over 0.8 and obtain at least 5 points for the preparation of the calibration graph (as usually recommended). Using 6 micropumps, as proposed in the manuscript, it is possible to obtain 7 concentrations to run a calibration graph. In addition, these points are evenly spaced on the graph.

The pumps were activated in a specific order to obtain the appropriate dilution of the standard used. The injection was programmed so that each injected portion of the standard with the diluent had a total nominal volume of 80 µL, which meant that 3 pumps were activated simultaneously. Table 4 shows the procedure of using the micro-pumps to obtain the calibration graph consisting of seven points. The nominal and real dilution, resulting from the true volume of micro-pumps, calculated according to the experimental data (see Section 2.1) are indicated.

To obtain an appropriate standard dilution, the micro-pumps indicated in Table 4 injected the liquids into the mixer at least 17 times, using about 1360 µL of the solution. This value was obtained through optimization and was matched to the length of flow lines used in the system. With the developed pump work program, it was possible for the solutions to reach the mixer and move to the ion-selective electrode. Then, during 60 s of stop-flow, the voltage was stabilized and read. Figure 3 shows an editing window of the computer program used. The example of the micro-pump sequence program for five calibration solutions preparation and for sample solution injection is presented. The voltage signal generated in the electrode is indicated by the red line.

It was calculated that the standard solution consumption in one calibration cycle was about 3400 µL, consumption of sample: 850 µL. Therefore, the total production of waste was about 7650 µL. Calibration throughput was at 7.8 cycles per hour^−1^ (cycle time of 460 s). However, considering the fact that the ISE potential stabilized quite quickly, it is possible to significantly shorten the measuring cycle time. The time to measure potential can be shortened to about 20 s. As a result, the measuring cycle time can be shortened to 220 s, which results in the throughput of the method of about 16 cycles per hour^−1^. In the presented research, the calibration cycle was repeated at least five times for us to be able to construct the calibration graph.

#### 3.2.3. Reference Method

Photometric determination of nitrate using cuvette test of Hach (LCK 339) was applied as a reference method. The determinations made with these tests are based on the dimethylphenol method. Nitrate ions in solutions that contains sulfuric and phosphoric acids react with 2,6-dimethylphenol to form 4-nitro-2,6-dimethylphenol. The measurement wavelength is 345 nm. The time of a single analysis takes more than 15 min. The tests were intended for the determination of nitrates in the range of 0.23–13.5 mg L^−1^ (NO_3_-N) or 1.0–60.0 mg L^−1^ NO_3_. The tests comply with standards: ISO 7890-1-2-1986, DIN 38405 D9-2. The DR3900 Laboratory VIS Spectrophotometer (Hach Company, Loveland, CO, USA) was used for this experiment.

## 4. Conclusions

The simple, fully automated flow system for the determination of nitrate in surface waters has been described. Thanks to the use of a system of solenoid micro-pumps, high precision of determinations was achieved. The recalibration of the micropumps led to improved accuracy of dosing solutions. Furthermore, no reagents are required for the nitrate determination, and only a small amount of relatively safe reagents (ISA- ammonium sulfate) are used for the calibration process. Although, when applying this method in practice, one should bear in mind that the reliability of the obtained results depends on the quality of the ion-selective electrode used, i.e., on its V = f(log CNO_3_) characteristic, which ceases to be linear in the area of low concentrations.

The developed instrumentation and method of calibration have been applied successfully for the determination of nitrate in surface waters. By analyzing the results, it was shown that compatibility with the reference cuvette test method is sufficient, but with a much shorter time of analysis. The obtained results were consistent with the values declared in the Certified Reference Material (CRM). The high throughput of the method and the ability to fully automate the detection process means that it is suitable for fixed-site water quality monitoring.

## Figures and Tables

**Figure 1 molecules-28-00493-f001:**
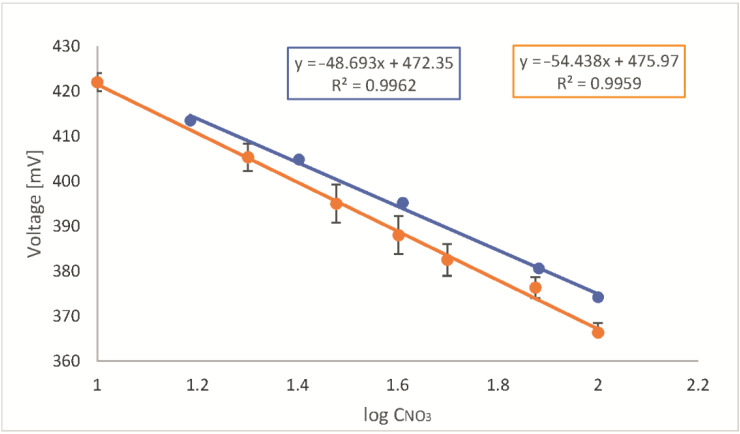
Examples of calibration graphs for NO_3_^−^ ions determination obtained in presented flow system (blue color) and in traditional, batch condition (yellow color); x-axis: Logarithm of the concentration expressed as mg L^−1^. Each point represents an average of at least three consecutive measurements.

**Figure 2 molecules-28-00493-f002:**
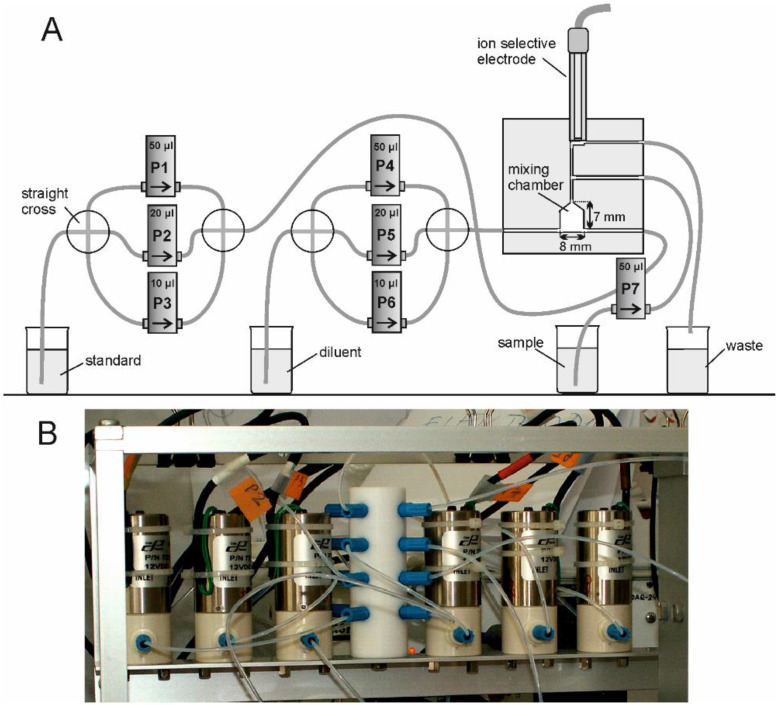
Schematic diagram of the applying flow network (**A**) and a photograph of micro-pumps P1 ÷ P6 with four straight crosses integrated into one Teflon block (**B**).

**Figure 3 molecules-28-00493-f003:**
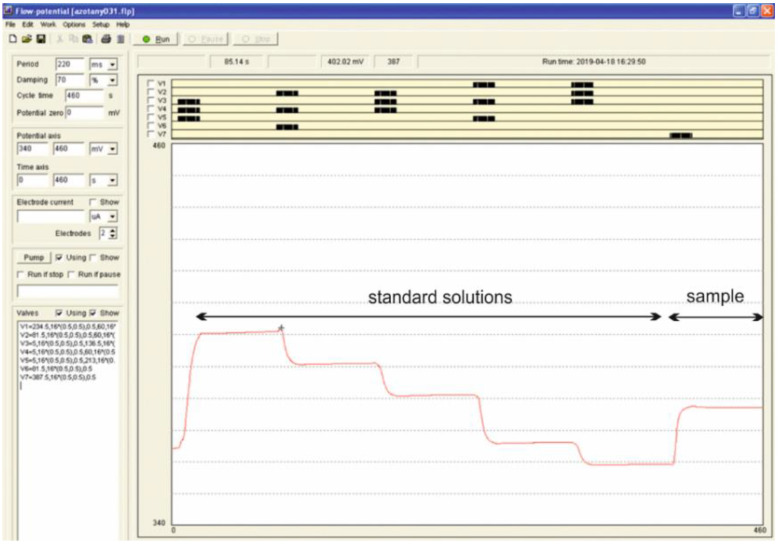
Editing window of the computer program used in this research. Program of micro-pumps for one measuring cycle (top) and corresponding voltage signal of ISE as a function of time (red line-bottom).

**Table 1 molecules-28-00493-t001:** Results of the re-calibration of the micro-pumps.

Micro-Pump	Nominal Volume [µL]	Mean, Real Stroke Volume [µL]	Precision (R.S.D. [%] ^1^)	Accuracy (R.D. [%] ^2^)
P1	50	54.3 (55.2)	0.11 (0.03)	8.6 (10.4)
P2	20	22.2 (22.7)	0.54 (0.20)	11.0 (13.5)
P3	10	13.1 (13.3)	0.91 (0.16)	31.0 (33.0)
P4	50	51.3 (52.2)	0.02 (0.09)	2.7 (4.4)
P5	20	21.0 (21.6)	0.38 (0.13)	5.2 (8.0)
P6	10	14.3 (14.8)	0.21 (0.10)	43 (48)
P7	50	52.3 (52.6)	0.27 (0.03)	4.7 (5.2)

^1^ Relative standard deviation for real stroke volume. ^2^ Relative deviation of real volume from nominal volume.

**Table 2 molecules-28-00493-t002:** Data on checking the calibration graphs.

Standard Concentration		Dilution Degree	Standard Concentration after Dilution log CNO_3_	ISE Voltage V [mv]	Sample Concentration log CNO_3_		Accuracy (R.D. [%])
[mg L^–1^]	log CNO_3_				Nominal	Found	
10	1.00	0.153	0.186	439.4 ± 0.6	0.70	0.65	−7.1
		0.253	0.403	438.0 ± 0.4			
		0.408	0.610	433.2 ± 0.2			
		0.762	0.882	421.9 ± 0.1			
		1.00	1.00	415.0 ± 0.1			
50	1.70	0.153	0.88	424.9 ± 0.7	1.00 1.30	0.97 1.27	−3.0 −2.3
		0.253	1.10	418.6 ± 0.5			
		0.408	1.31	410.0 ± 0.3			
		0.762	1.58	396.5 ± 0.5			
		1.00	1.70	389.9 ± 0.3			
100	2.00	0.153	1.19	413.4 ± 0.6	1.00 1.30	0.99 1.26	−1.0 −3.1
		0.253	1.40	404.8 ± 0.4			
		0.408	1.61	395.2 ± 0.2			
		0.762	1.88	380.6 ± 0.2			
		1.00	2.00	374.2 ± 0.2			
1000	3.00	0.153	2.19	359.5 ± 0.2	2.30 2.70	2.28 2.68	−0.9 −0.7
		0.253	2.40	349.6 ± 0.1			
		0.408	2.61	339.4 ± 0.1			
		0.762	2.88	324.6 ± 0.1			
		1.00	3.00	318.12 ± 0.1			

**Table 3 molecules-28-00493-t003:** Results of determination of log CNO_3_^−^ ions concentration in lake and pond water. The results represent the average of at least four determinations ± S.D.

Sample	ISE		Cuvette Test
	Proposed Flow System	Manual Procedure	
Lake water-1	1.43 ± 0.02	1.41	1.36 ± 0.00
Lake water-2	1.37 ± 0.01	1.36	1.30 ± 0.01
Pond water-1	0.55 ± 0.01	0.56	0.34 ± 0.04
Pond water-2	0.56 ± 0.01	0.60	0.33 ± 0.01

**Table 4 molecules-28-00493-t004:** Procedure of standard dilution during the calibration step.

Point	Pumps Activated	Volume Ratio (Standard/Total)	Nominal Standard Dilution	Real Standard Dilution ^1^
1.	P3 + P4 + P5	$\frac{10}{10+50+20}$	0.125	0.153
2.	P2 + P4 + P6	$\frac{20}{20+50+10}$	0.250	0.253
3.	P2 + P3 + P4	$\frac{20+10}{20+10+50}$	0.375	0.408
4.	P1 + P5 + P6	$\frac{50}{50+20+10}$	0.625	0.606
5.	P1 + P3 + P5	$\frac{50+10}{50+10+20}$	0.750	0.762
6.	P1 + P2 + P6	$\frac{50+20}{50+20+10}$	0.850	0.843
7.	P1 + P2 + P3	$\frac{50+20+10}{50+20+10}$	1	1

^1^ Dilution after taking into account the true volume of the micro-pumps.

## Data Availability

Not applicable.

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
