# Peer review of "A Multi-Pumping Gradient Calibration Module for Potentiometric Determination of Nitrate in Surface Water"

_molecules, 2023, doi:10.3390/molecules28020493_

Round 1

Reviewer 1 Report

The manuscript of S. Koronkiewicz and colaborators describes a continuous flow-system for potentiometric nitrate analysis in waters using seven micropumps with different stroke volumes for in-line preparation of calibrating solutions. The manuscript is written in understandable English and very clear regarding the main goals and the experiments rationale. I do not think, nevertheless that authors have looked for a comprehensive knowledge on issues raised by the use of potentiometric detection, and hence fully exploited the capabilities of their proposal. I strongly recommend the reading of a reference paper in J. Pharm. Biomed. Anal. 25(2001) 713-720. In it a full-automatic system providing off-line and in-line calibration, evaluation of interferences by the fixed and separated solutions methods, direct measurement of samples, standard addition and titration was proposed.  On other hand the analytical proposal is far from being validated, regarding interferences and reproducibility data.

A third issue regards on how can an interested reader reproduce the system from a simple picture of the pumps (Fig 3). The text has no details how they are driven both by hardware and software.  How the number of micropumps was established? How the nitrate stock solution is changed? What is the volume of the mixing chamber? Why micropump 7 is placed just before the electrode? How bubbles accumulation everytime the sample is changed was tackled?

A common referred issue on the use of micropumps is the volumes delivered to the manifold are strongly dependent on the backpressure imposed by this. Data in Table 1 should be completed with data obtained after placing 1 or 2m long tubing after the micropumps. Maybe that this would explain why all found concentrations in Table 2 are lower than the predicted values.

The conventional calibration of nitrate electrode should be added for comparison of slopes and differences should be critically evaluated.

In table III, all values got with the proposal are higher than the reference cuvette test; why it is so?

Reviewer 2 Report

This paper deals with the development of a multi pumping system for the potentiometric determination of nitrate. The main novelty is the preparation of the standard solutions in the flow system.

This paper is well written, the results are well presented and seem to be sound, and the flow system has potential as a proof of concept, and so I recommend publication with some revision. However, the following points should be addressed by the authors, as a major revision:

1)    The authors call this method a flow system; I have some doubts. Although pumps are used, the measurement is carried out in non-flowing stopped-flow batch conditions; maybe it should be named a flow-batch system, or then stopped-flow, like others in the literature.

2)    Why didn´t the authors use another pump to adjust the ionic strength of standards and samples? Apparently, it is carried out off-line, which implies a unnecessary sample treatment.

3)    The authors should present a comparison of the results obtained with those obtained by the reference photometric method; the authors refer this method but I found no comparison of results.

Reviewer 3 Report

In this manuscript, a multi-pumping flow system used for different dilution of the standard was described as a way for automatic on-line calibration. The applicability of the system was confirmed by ISE determination of nitrate. Repeatable results were obtained and the accuracy of the method was checked through real samples. The manuscript is clear written. I support its publication in Molecules and there are only a couple of minor issues that may need to be considered:

1. Re-calibration of the micropumps is a good measure to ensure the accuracy of the dilution, but is there any change of the volume after a long period of operation, such as a month? How much the effect of the ambient temperature on these volumes?

2. For the nitrate determination, how many is the least points of the calibration concentrations?

3. Error bars should be added in Figure 1.

Round 2

Reviewer 1 Report

Authors had reasonably addressed the main issues and suggestions and, in accordance, I have no further objections to the acceptance of the improved revised version

Reviewer 2 Report

Publish as it is.